# Comparison between Novice and Experienced Surgeons Performing Corrective Osteotomy with Patient-Specific Guides in Dogs Based on Resulting Position Accuracy

**DOI:** 10.3390/vetsci8030040

**Published:** 2021-02-28

**Authors:** Yoon Ho Roh, Cheong Woon Cho, Chang Hun Ryu, Je Hun Lee, Seong Mok Jeong, Hae Beom Lee

**Affiliations:** College of Veterinary Medicine, Chungnam National University, Daejeon 34134, Korea; royoonseok@gmail.com (Y.H.R.); bluevet627@gmail.com (C.W.C.); modd3@naver.com (C.H.R.); jehunlee@hanmail.net (J.H.L.); jsmok@cnu.ac.kr (S.M.J.)

**Keywords:** 3D printing, angular limb deformity, surgical guide, dog

## Abstract

Corrective osteotomy has been applied to realign and stabilize the bones of dogs with lameness. However, corrective osteotomy for angular deformities requires substantial surgical experience for planning and performing accurate osteotomy. Three-dimensional printed patient-specific guides (3D-PSGs) were developed to overcome perioperative difficulties. In addition, novices can easily use these guides for performing accurate corrective osteotomy. We compared the postoperative results of corrective osteotomy accuracy when using 3D-PSGs in dogs between novice and experienced surgeons. We included eight dogs who underwent corrective osteotomy: three angular deformities of the radius and ulna, three distal femoral osteotomies, one center of rotational angle-based leveling osteotomy, and one corrective osteotomy with stifle arthrodesis. All processes, including 3D bone modeling, production of PSGs, and rehearsal surgery were carried out with computer-aided design software and a 3D-printed bone model. Pre- and postoperative positions following 3D reconstruction were evaluated by radiographs using the 2D/3D registration technique. All patients showed clinical improvement with satisfactory alignment and position. Postoperative accuracy evaluation revealed no significant difference between novice and experienced surgeons. PSGs are thought to be useful for novice surgeons to accurately perform corrective osteotomy in dogs without complications.

## 1. Introduction

Corrective osteotomy in veterinary medicine is primarily applied to correct bone malformations due to medial patella luxation with varus or valgus deformity of the femur or angular limb deformities [1]. Early surgical treatment of these problems can help to decrease the progression of osteoarthritis [2,3]. Corrective osteotomy, a procedure involving cutting of the bone for the purpose of appropriate alignment, is a complex technique. Conventional plans involve radiography and two-dimensional (2D) computed tomography (CT), and elective surgery that realigns and stabilizes the bone by cutting the metaphysis or diaphysis is needed [1,2]. However, the osteotomy position and reduction during surgery is primarily based on decisions made by the surgeon. The success of these procedures is highly dependent on precise pre-operative planning and substantial orthopedic surgery experience [4,5,6]. Thus, corrective osteotomy is a challenging procedure for a novice surgeon with limited experience.

Three-dimensional (3D) computer-aided design (CAD) software and 3D-printed patient-specific guides have been developed based on recent advances in technology [7,8,9,10,11]. Three-dimensional printed patient-specific guides (PSGs) are specially designed guides for a patient’s bone anatomy for use in osteotomy and reduction procedures during surgery. Several reports have recently described the use of 3D-printed patient-specific guides (3D-PSGs) in addressing fractures of various sites, such as the cervical spine, tibia, radius, and scapula as well as in corrective veterinary surgery [7,8,12,13,14,15]. The introduction of these techniques has led to a reduction in operation time, minimization of surgical incision, and improved accuracy in corrective osteotomy [6,8,13,16,17]. It is reported that the 3D-printed guide allows rapid, accurate, and safe placement of screws in the bone of the patient, thereby enhancing stability [14,17]. In addition, the PSGs can be readily used by novices to achieve successful outcomes. PSGs can specify the location, range, and degree of manipulation, including during osteotomy and reduction during surgery, without relying on a ruler and anatomic variation based on the preoperative plans [8,9,13,16].

However, there have been a few reports regarding these techniques in veterinary medicine. In addition, these reports have the limitation of a small number of cases to compare. The purpose of this study is to compare the outcomes between novice and experienced surgeons in terms of the position and alignment accuracy after corrective osteotomy using 3D-PSGs. We also evaluated risk factors and disadvantages of the technique.

## 2. Materials and Methods

### 2.1. Data Collection

All patients who underwent a corrective osteotomy with 3D-PSGs for the treatment of bone deformities between 1 January 2018 to 1 June 2020 were eligible for inclusion. Data collected from the medical records included patient signalment, body weight, history, radiographic and computed tomographic imaging, and type of bone deformities. All patients who underwent a preoperative CT scan and were evaluated postoperatively after a minimum follow-up of six months were included. The occurrence of complications was divided into minor or major. Minor complications were considered as those resolved without additional surgery (dehiscence, incisional irritation, and seroma). Major complications were determined as those needing additional surgery (infection, non- and malunion, and removal of implants) [18,19].

The identity of the primary surgeon was recorded, and a note was made regarding whether he or she was a faculty surgeon who had performed more than 40 corrective osteotomies or a novice who had never performed one. Cases operated by the experienced surgeon were classified as the experienced group, whereas the other cases were classified as the novice group.

### 2.2. Guides Production and Surgery

Computed tomography (Alexion^TM^, Canon medical systems corporation, Otawara, Japan) images of the patient were obtained using bone and soft tissue filters, with operating parameters of 120 kV and 12 mA. Images were obtained with a slice thickness of 1 mm and reconstructed in computer software (3DS Max, Autodesk, CA, USA). All processing, from analysis of bone deformity to production of patient-specific guides, was carried out using 3DS Max. Bone models were measured and made according to a previously described protocol [10]. A cutting guide and reduction guide were created to provide an accurate osteotomy line for plane and press-fit during reduction, respectively. These 3D guides were designed to fit to the anatomic features of the bone, such as protuberances. To ensure a precise and appropriate fit of the drill guide to the bone surface, its ventral surface was designed to represent an inverted virtual representation of the dorsal cortex of the bone. In addition, drill or k-wire guide sleeves were created to ensure a safe pathway for the screw and k-wire [17]. The bone model and PSGs (Figure 1) were printed using a 3D printer (Finbot-Z420, TPC Mecatronics, Seoul, Korea) and polylactic acid filament (PLA filament, Cubicon, Seongnam, Korea). All novice and experienced surgeons performed rehearsal surgery before the operation to practice the plans that were predeveloped in computer software.

The patient was pre-medicated with midazolam and remifentanil constant rate infusion (CRI). Anesthesia was induced with propofol and maintained with isoflurane. Cefazolin was administered 30 min prior to incision and every 90 min during the surgery. Surgeries were performed with the 3D-printed PSGs, which were sterilized using ethylene oxide gas before operation.

### 2.3. Postoperative Evaluation of Position and Outcome

Postoperative errors were evaluated by comparing the position of bone segments in a computer program (3DS Max, Autodesk, CA, USA) using the 2D/3D registration technique (Figure 2). They were expressed with six degrees of freedom on the basis of the local coordinate axes in each patient (Figure 2). In that program, the result of the computer-assisted 3D-rehearsal surgery bone model was overlapped with postoperative radiographic images (which were imported into the program) on the same plane to evaluate the difference in the position of bone segments. Based on the osteotomy line, the proximal segment of the computer-assisted 3D-rehearsal surgery bone model was rotated and translated to match the proximal segment of the postoperative radiograph image. The distal segment, denoted as distal segment 1, was also rotated and translated spontaneously, because the distal segment is attached to the proximal segment. Concurrently, another distal segment of the 3D-rehearsal surgery bone model, denoted as distal segment 2, was manipulated to match the distal segment of the postoperative radiographic image. The distribution of the spatial relationship of distal segment 1 and distal segment 2 was compared to measure the degree of translation/rotation. Every bone model was analyzed according to the method mentioned above. Raw data of errors were described using means and standard deviations and showed proper limb alignment and implant positioning in all cases.

A visual analog scale questionnaire consisting of 12 questions was performed directly before and after surgery by the owner (data not shown) [20,21]. The postoperative response was at least six months.

### 2.4. Statistical Analysis

The statistical analyses were performed using SPSS version 26.0 software (SPSS Inc., Chicago, IL, USA). Errors of translation and rotation were compared between the experienced and novice group. The Mann–Whitney test was used to compare significant differences between the groups. Significance was set at *p*  <  0.05.

## 3. Results

### 3.1. Population Data

A total of eight dogs underwent surgery using the 3D-printed patient-specific surgical guides. An experienced surgeon operated on three dogs and a novice surgeon on five dogs (Table 1). There were no intraoperative complications in any case. Breed distribution was as follows: Chihuahua (*n* = 1), Golden Retriever (*n* = 1), Maltese (*n* = 1), Miniature Pinscher (*n* = 1), Pomeranian (*n* = 3), and Welsh Corgi (*n* = 1). The median age and body weight at the time of surgery were 2 years (range, 1–9 years) and 2.95 kg (range, 1.4–33 kg), respectively. Other descriptions are shown in Table 1. All cases required one week for surgical planning and printing of the bone model and PSGs. All surgeries were completed within 2 h regardless of the level of surgeon experience.

### 3.2. Surgical Details

Cases included three angular limb deformity corrections of the radius and ulnar, two distal femoral osteotomies (DFOs) with tibial tuberosity transposition, one DFO with cranial tibial closing wedge osteotomy (CTWO), one center of rotational angle-based leveling osteotomy (CBLO), and one corrective osteotomy with stifle arthrodesis (Figure 1). In case 1, deformity of radius and ulna was corrected by a monoapical open wedge considering the shortened leg. In cases 2, 3, 7, and 8, a biapical osteotomy was performed to correct angular limb deformities. In case 4, a femoral osteotomy (DFO) and biapical cranial tibial wedge osteotomy (CTWO) were performed to correct femoral varus, tibial valgus, and cranial cruciate ligament rupture. In case 5, to correct excessive tibial plateau angle (eTPA) and tibial varus, a biplanar closing center of rotational angle-based leveling osteotomy (CBLO) was performed.

### 3.3. Postoperative Assessment of Position and Outcome

The degree of translation/rotation of the postoperative radiographic image relative to the computer-assisted 3D rehearsal surgery bone model was evaluated (Table 2). The mean degrees of translation were 0.57, 1.04, and 0.68, and degrees of rotation were 0.86, 1.59, and 0.89, on the x, y, and z axes, respectively. There was no difference in accuracy between postoperative outcome and 3D rehearsal surgery in the computer software. The mean degrees of translation were 0.52, 1.05, and 0.61, and degrees of rotation were 0.71, 1.60, and 0.60 on the x, y, and z axes, respectively, in the novice surgeon group. The mean degrees of translation were 0.7, 1.01, and 0.89, and degrees of rotation were 1.3, 1.55, and 1.77, on the x, y, and z axes, respectively, in the experienced surgeon group. There were no significant differences between novice and experienced groups (*p* > 0.05). The use of PSGs provided good limb alignment and limb function.

All patients showed clinical improvement in lameness and satisfactory alignment and position without minor and major complications associated with malalignment or infection.

## 4. Discussion

The postoperative accuracy of PSGs was evaluated using the 2D/3D registration measurement technique. There was no significant difference in the assessment of the postoperative position of the bone between novice and experienced surgeons. Therefore, the results of the present study support that corrective osteotomy with patient-specific guides can be an effective treatment for even novice surgeons and achieve high owner satisfaction.

Small errors after corrective osteotomy may increase the risk of complications in dogs [2,22]. Several factors can affect the errors and results of surgery, namely, preoperative, intraoperative, and postoperative factors [16]. In this paper, the total average translation was about 0.57, 1.04, and 0.68 mm on the x, y, and z axes, respectively, and average rotation, 0.86, 1.59, and 0.89 mm on x, y, and z axes, respectively, when analyzing each deformity plan. It is likely that the amount of postoperative deviation found in this study did not affect the clinical outcome because all patients showed favorable prognoses without major complications. Although the errors found in this study (postoperative translation and rotation) appeared to be larger than those found in previous research [16], these discrepancies (mm) were minimal compared to the size of the bone. However, the cause of these errors could relate to the procedure while performing an osteotomy or reduction or when inserting a screw. The soft tissue surrounding the bone compromises the surgeon’s vision, and it is difficult to perceive the precise osteotomy line. In addition, bending of the saw and guide due to excessive power from the surgeon can cause postoperative errors. Although the bone loss from osteotomy in 3D rehearsal surgery was calculated as the thickness of a 0.6 mm saw blade (Stryker TPX, Stryker Inc., Kalamazoo, MI, USA), more significant loss of bone from poor handling technique can affect the outcome. However, there were no significant differences between novice and experienced groups (*p* > 0.05). One of the PSG advantages is that surgical outcome can be similar to a preoperative plan [8,16]. Therefore, these errors could result from procedures, including performing radiography and comparing the position of bone segments in a computer program.

The PSGs can be categorized as several types depending on their application method [11]. The advantage of PSGs is that various types can be applied according to the preference of the surgeon and patient size. The osteotomy guide determines the position of the saw desired by the surgeon without measuring the bone by ruler, increasing the accuracy of designating the osteotomy line and reducing the surgical time [8,15]. All surgeries in this study were completed within two hours, regardless of the experience of the surgeon. The 3D-PSG and rehearsal surgery with a bone model helped the surgeon identify the exact location for the osteotomy while allowing the surgeon to be familiar with the bone’s circumstances prior to surgery [9]. For reduction, distracting bone fragments in the patients with soft tissue contracture is challenging after osteotomy. In addition, maintaining the segments of the distracted bone with torsion or varus/valgus deformities is more difficult [11]. The reduction guide not only prevents excessive translation and rotation in the process of reduction after the osteotomy but also allows segments of bones to be positioned according to plans because the ventral surface of this reduction guide was modeled to completely match the dorsal surface of the postoperatively reduced bone on CAD surgery [8,15]. K-wires for the reduction guide were placed to the bone before making an osteotomy with the osteotomy guide. The position of the k-wire was planned based on the postoperative feature of reduced bone [17]. Therefore, as the reduction guide glided through the k-wires to contact the bone segment surface, it caused the bone segments to translate and rotate to optimal position. Compared to Jig, which is commonly used to reduce and stabilize in orthopedic surgery, the PSGs are more precise, simpler with less iatrogenic bone damage [16]. They can reduce the risk of postoperative bone fracture in dogs. Although a significant amount of time is required to prepare PSGs and bone models for rehearsal surgery, this is compensated by improvements in the accuracy of postoperative outcome and shorter operating hours.

This study has some limitations, including the small sample size of dogs receiving treatment; it is possible that the surgical outcome would be different with a larger sample size. Although each case has different guides and different surgical procedures, various types of PSGs could be applied to diverse cases due to their natural and specific features. Moreover, the purpose of this study was not to provide guidelines about the types of PSG that should be clinically indicated in dogs with bone deformities. In addition, the issues of cost and the time for the production of the PSGs must be considered. Further studies are needed to evaluate the efficacy of this technique with regard to complete resolution and the cost of long-term follow-up.

PSGs can be beneficial in surgery, despite the varieties of dog breeds and techniques. Although the precise methods and systems for the use of PSGs have not been conclusively determined, considering the related studies and technological development, the PSG system is thought to be useful for corrective osteotomy on dogs, particularly for novice surgeons who intend to improve their accuracy of surgical correction in veterinary medicine. Based on our experience, the application of PSGs appears to be feasible, resulting in excellent accuracy and limited postoperative complications.

## Figures and Tables

**Figure 1 vetsci-08-00040-f001:**
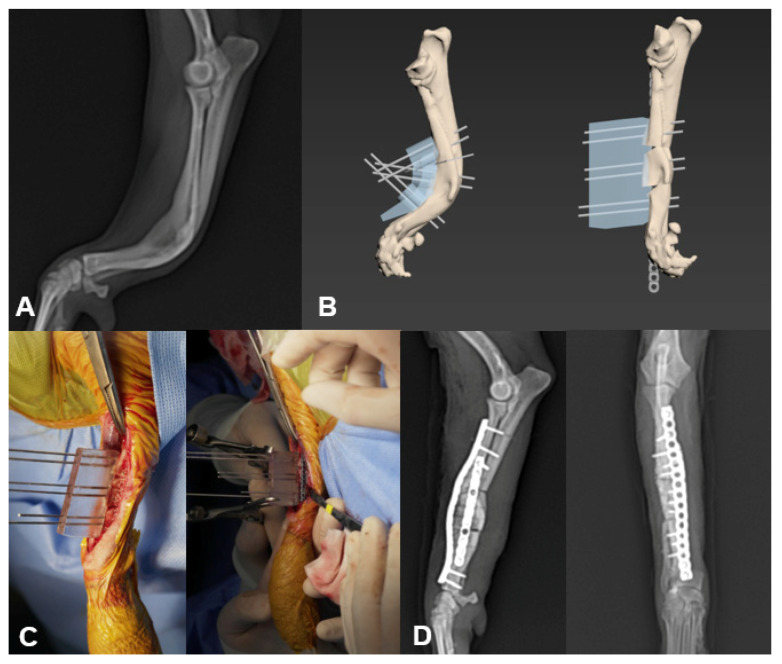
The application of patient-specific guides in case 1. Preoperative mediolateral radio-ulnar projection (**A**). Rehearsal surgery in 3D computer software with osteotomy guide and reduction guide (**B**). K-wires were accurately inserted in the canal of a patient-specific guide (PSG) connected with the bone before osteotomy, and the segments of bone were translated and rotated to an optimal position with the reduction guide. Intraoperative application of PSG (**C**). Plates were precontoured on the plan to avoid K-wires. Immediate postoperative mediolateral and craniocaudal radiographs of the affected limb (**D**). PSGs, patient-specific guides.

**Figure 2 vetsci-08-00040-f002:**
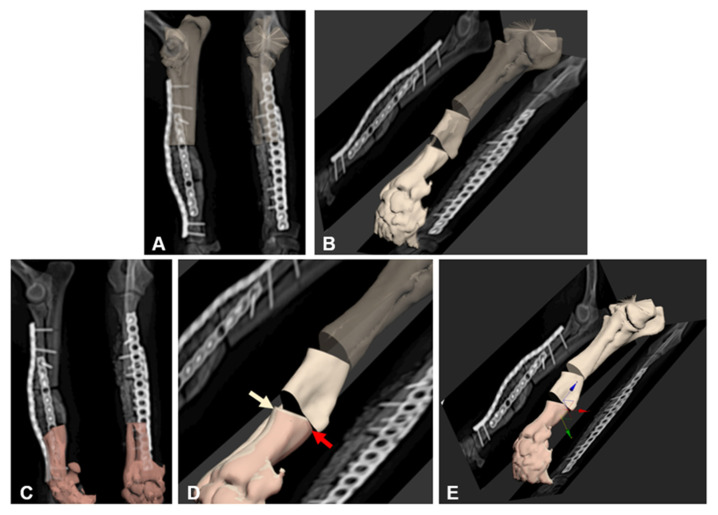
The method for measuring errors between preoperative plan and surgical outcome in computer-aided design (CAD). The proximal segment of the synthetic projection view of preoperative 3D bone images is overlaid with the same part of the postoperative radiograph of the bone in the same plane (**A**). Based on the relative positions in CAD, other bone segments were placed (**B**). In the same manner as in (**A**), one of the other bone segments was overlaid with the same part of the postoperative radiograph on the radiograph plane (**C**). Errors between the preoperative plan (beige arrow) and the surgical outcome (red arrow) were then measured (**D**). The degree of translation/rotation was measured on the x, y, z axes (**E**).

**Table 1 vetsci-08-00040-t001:** Signalment, diagnosis, surgery, surgeon proficiency, and type of PSG in eight dogs undergoing surgical correction by 3D-printed patient-specific guides.

Number	Group	Signalment	Diagnosis	Type of Surgery	Surgeon Proficiency	Type of PSG
1	E	A 2-yr-old, 1.9 kg, F, Pomeranian	ALD of radius and ulnar with valgus and recurvatum	Monoapical open wedge osteotomy	Experienced	Osteotomy/Reduction
2	N	A 2-yr-old, 2.5 kg, F, Pomeranian	ALD of radius and ulnar with recurvatum and external torsion	Biapical neutral wedge osteotomy	Novice	Osteotomy/Reduction
3	N	A 1-yr-old, 14 kg, CM, Welsh Corgi	ALD of radius and ulnar with procurvatum and external torsion	Biapical closing wedge osteotomy	Novice	Osteotomy/Reduction
4	N	A 6-yr-old, 3.4 kg, CM, Maltese	Bilateral MPL and CCLR with tibial valgus	DFO and Biapical CTWO	Novice	Osteotomy/Reduction
5	E	A 2-yr-old, 33 kg, CM, Golden Retriever	Bilateral MPL and CCLR with tibial varus	Biplanar CBLO	Experienced	Osteotomy/Reduction
6	E	A 9-yr-old, 2.5 kg, SF, Pomeranian	Patella tendon rupture and quadriceps contracture	Corrective osteotomy with Stifle arthrodesis	Experienced	Osteotomy/Reduction
7	N	a 2-yr-old, 2.8 kg, SF, Maltese	Bilateral MPL	DFO	Novice	Osteotomy/Reduction
8	N	a 4-yr-old, 4.1 kg, M, Chihuahua	Left MPL	DFO	Novice	Osteotomy/Reduction

ALD, angular limb deformity; CCLR, cranial cruciate ligament rupture; CM, castrated male; CTWO, cranial tibial closing wedge; DFO, distal femoral osteotomy; E, experienced group; F, female; M, male; MPL, medial patella luxation; N, novice group; SF, spayed female; yr, year.

**Table 2 vetsci-08-00040-t002:** Raw data of errors of translation and rotation in corrective osteotomy of eight dogs.

Case	Group	Translation (mm)			Rotation (mm)		
		X	Y	Z	X	Y	Z
1	E	1.61	0.44	2.04	0.06	1.96	4.67
2	N	0.69	0.40	0.68	0.38	1.62	0.01
3	N	0.47	0.24	1.26	0.09	0.71	0.15
4 (femur)	N	0.61	0.37	0.51	2.06	2.31	1.23
4 (tibia)	N	0.70	0.73	0.13	2.13	1.66	3.99
5	E	0.12	0.09	0.35	2.66	2.55	0.46
6	E	0.36	2.50	0.27	1.18	0.13	0.18
7 (Lt femur)	N	0.40	0.60	0.15	0.20	1.60	0.00
7 (Lt tibia)	N	0.00	1.10	0.70	0.38	1.62	0.01
7 (Rt Femur)	N	0.50	0.00	0.60	0.38	1.62	0.01
7 (Rt tibia)	N	0.82	5.00	0.80	0.38	1.62	0.01
8	N	0.50	1.00	0.70	0.38	1.62	0.01
Mean/SD	Total	0.57/0.4	1.04/1.41	0.68/0.53	0.86/0.91	1.59/0.64	0.89/1.65
Mean/SD	E	0.7/0.8	1.01/1.3	0.89/1	1.3/1.3	1.55/1.26	1.77/2.52
Mean/SD	N	0.52/0.24	1.05/1.52	0.61/0.34	0.71/0.79	1.6/0.4	0.6/1.33
*p*-value		0.456	0.368	0.659	0.456	0.22	0.38

E, experienced group; N, novice group; SD, standard deviation; X, direction medial to lateral; Y, cranial to caudal; Z, proximal to distal in both translation and rotation.

## Data Availability

Data available on request due to restrictions eg privacy or ethical.

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
