# Peer review of "Comparison between Novice and Experienced Surgeons Performing Corrective Osteotomy with Patient-Specific Guides in Dogs Based on Resulting Position Accuracy"

_vetsci, 2021, doi:10.3390/vetsci8030040_

Round 1

Reviewer 1 Report

The title is suitable for the article.

General comments are listed:

Line 28: Keywords should not be the same words of the title: in this case there is the repetition of corrective osteotomy and patient-specific guides.

Line 34: the orthopedic conditions listed above lead to osteoarthritis (OA) even if a surgery is performed, because OA is a chronic and degenerative process that cannot be stop by a surgery.

Line 48: What do the Authors mean with “cervix”? Maybe they should rather refer to cervical spine.

Line 70: It would be desirable a reference for the classification into minor or major complications: for example Cook et al. (2010) proposed definitions and criteria for reporting time frame, outcome and complications for clinical orthopedic studies in Veterinary Medicine. This indicates a lack of study of veterinary literature.

Line 89 to 99: The criteria according to whom a surgeon was classified as novice or experienced one is not clear, Authors should describe it.

Line 137: the title refers to small-breed dogs, among the enrolled dogs there was a golden retriever dog, so not so small (2 years and 33kg of weight!).

Line 139: In brackets, it would be better to indicate that the age range is expressed in years.

Table 1: so the surgeries were not equally divided between experienced and novice surgeons ( 3 in the first case, 5 in the second).

PAGE 6-7-8: where are the line numbers? There is a formatting error.

Page 6 Short-term outcome: how did the Authors evaluate it? How did they evaluate lameness degree before and after surgery and how did they compare them? It is not clear.

PAGE 7 discussion: how did the Authors evaluate the owner satisfaction? Did they use a questionnaire? If so, how was it organized?

Were there any minor complications? It is not clear.

There are some typing errors, but it difficult to show you the line because there are no numbers.

References n. 16 and 18 refers to the placement of pedicle screws and vertebral wall; in the cases reported in this paper the bones are different.

The bending of the saw is an error which is more common for novice surgeons.

The level of English is at accepted level.

The article is interesting, a correct preoperative planning is mandatory to achieve a good goal and the use of a 3D model and guide could be a good aid. Anyway, it needs some revisions.

Author Response

Response to reviewers' comments

Dear Reviewers

We want to thank the reviewers and editor for their thoughtful review of the manuscript. They raise important issues, and their comments have been constructive for improving the manuscript. We hope that the reviewers will find our responses satisfactory. We are willing to further edit the revised version of the manuscript in response to any additional suggestions they may have.

Response to comments from Review 1

COMMENT: Line 28: Keywords should not be the same words of the title: in this case there is the repetition of corrective osteotomy and patient-specific guides.
RESPONSE: As suggested, this change has been made (Line 28).

COMMENT: Line 34: the orthopedic conditions listed above lead to osteoarthritis (OA) even if a surgery is performed, because OA is a chronic and degenerative process that cannot be stop by a surgery.
RESPONSE: We thank the reviewer for commenting on important issues related to our study. Previous studies indicate that osteoarthritis continues despite the surgical treatment. But TPLO has been reported to prevent the progression of secondary joint disease and osteoarthritis (Vasseur et al 1992). Thus, we revised sentences as follows (Line 33 to 34)

COMMENT: Line 48: What do the Authors mean with "cervix"? Maybe they should rather refer to cervical spine.
RESPONSE: As suggested, this change has been made (Line 48).

COMMENT: Line 70: It would be desirable a reference for the classification into minor or major complications: for example Cook et al. (2010) proposed definitions and criteria for reporting time frame, outcome and complications for clinical orthopedic studies in Veterinary Medicine. This indicates a lack of study of veterinary literature.
RESPONSE: As suggested, this change has been made (Line 74).

COMMENT: Line 89 to 99: The criteria according to whom a surgeon was classified as novice or experienced one is not clear, Authors should describe it.
RESPONSE: We would like to thank the reviewer for commenting on important issues that could be misleading. Thus, we revised these sentences as follows (Line 74 to 78)

â–¶The identity of the primary surgeon was recorded, and a note was made regarding whether they were a faculty surgeon who has performed more than 20 corrective osteotomies or a novice who has never performed it. Cases operated by the experienced surgeon were classified as the experienced group, whereas the other cases were classified as the novice group.

COMMENT: Line 137: the title refers to small-breed dogs, among the enrolled dogs there was a golden retriever dog, so not so small (2 years and 33kg of weight!).
RESPONSE: We would like to thank the reviewer for commenting on important issues that could be misleading. Thus, we changed "small-breed dogs" to "dogs" in this paper, including the title.

COMMENT: Line 139: In brackets, it would be better to indicate that the age range is expressed in years.
RESPONSE: As suggested, this change has been made (Line 152).

COMMENT: Table 1: so the surgeries were not equally divided between experienced and novice surgeons (3 in the first case, 5 in the second).
RESPONSE: We thank the Associate Editor for commenting on important issues related to our study. We agree with the reviewer's comments that there is not a statistical analysis in your material and methods selection, so you can't draw and support your results objectively.

In the process of considering materials and methods, we failed to get more patients with corrective osteotomy. There are statistical analysis concerns about comparing the differences between the groups because of the small sample size. Thus, we decided to add the statistical analysis using the Mann-Whitney test and elucidate the limitation of comparability of this study as follows (Line 139 to 143)

â–¶Table 1 and 2.

â–¶2.4. Statistical analysis

The statistical analyses were performed using SPSS version 26.0 software (SPSS Inc., Chicago, IL, USA). Errors of translation and rotation were compared between experienced and novice group. Mann-Whitney test was used to compare significant differences between the groups. Significance was set at p < 0.05.

COMMENT: PAGE 6-7-8: where are the line numbers? There is a formatting error.

There are some typing errors, but it difficult to show you the line because there are no numbers.
RESPONSE: As suggested, lines have been inserted on the page 6-7-8.

COMMENT: Page 6 Short-term outcome: how did the Authors evaluate it? How did they evaluate lameness degree before and after surgery and how did they compare them? It is not clear. PAGE 7 discussion: how did the Authors evaluate the owner satisfaction? Did they use a questionnaire? If so, how was it organized
RESPONSE: We agree with Associate Editor's comment that more information is needed to explain outcome evaluation. We used a visual analog scale questionnaire consisting of 12 questions. But the data related to lameness evaluation was not shown in this paper because this focused on the postoperative position accuracy of bone. Thus, we decided to add the text as follows (Line 126 to 128) and edited the sentences as follows (Line 127 to 129) and (Line 189 to 192)

â–¶ A visual analog scale questionnaire consisting of 12 questions was performed directly before and after surgery by the owner (data not shown) [19,20]. The postoperative response was at least 6 months.

â–¶ All patients showed clinical improvement in lameness and satisfactory alignment and position with no major complications associated with malalignment or infection. In addition, all owners were satisfied that the result was without major complications, such as requiring additional surgery.

COMMENT: Were there any minor complications? It is not clear.
RESPONSE: All patients showed postoperative satisfactory alignment and position without minor and major complications associated with malalignment or infection. This change has been made as follows (Line 189 to 192).

COMMENT: References n. 16 and 18 refers to the placement of pedicle screws and vertebral wall; in the cases reported in this paper the bones are different.
RESPONSE: We thank the reviewer for highlighting this critical issue. As we mentioned previously, this study is designed to focus on the evaluation of postoperative position accuracy of bone. The reference n. 16 reported the effectiveness and accuracy of the 3D guide for designating the placement of screws and drilling in the postoperative assessment. 3D guides used in this study also played a similar role in our surgeries. Therefore, we revised the texts that could be misleading and delete the reference n. 18 as follows (Line 211 to 215)

â–¶ Although the errors found in this study (postoperative translation and rotation) appear to be larger than those found in previous research [15], these discrepancies (mm) were min-imal compared to the size of the bone. However, the cause of these errors could relate to the procedure while performing an osteotomy or reduction or when inserting a screw.

COMMENT: The bending of the saw is an error which is more common for novice surgeons..
RESPONSE: We agree to the reviewer's comments on the important issues that could be misleading. We decided to edit repeated sentences and add new text about PSGs as follows (Line 216 to 224). We totally organized the discussion section

Reviewer 2 Report

I would thank the authors for they work. I think that the use of PSGs is very interesting, but I have some serious doubts about this paper.

I would suggest to revise accurately the grammary of the paper.

The title is not correct: you speak about the use of PSGs in small animal but in your cases one of the patient is a Golden retriver.

You speak about the use of PSGs to perfrom corrective osteotomy, but in your paper it is included a stifle artrodesis.

In the title and in all the paper you say that the object of the study is to compare results of novice and experienced surgeon in performing corrective osteotomy,but there is not a statistical analysis in your material and methods selection, so you can't draw and support objectively your results.

In the title and in all the paper you say that the object of the study is to compare results of novice and experienced surgeon in performing corrective osteotomy,but in authors contribution there is written that HBL performed the corrective surgery for all dogs. So I do not understand how it is possibile to compare the accuracy between novice and experienced surgeons if all the procedures were carried out by the same surgeon.

Discussion section is quite low of content, with the same concepts repeated more times but not a true and organized discussion has been carried out.

Author Response

Response to reviewers' comments

Dear Reviewers

We want to thank the reviewers and editor for their thoughtful review of the manuscript. They raise important issues, and their comments have been constructive for improving the manuscript. We hope that the reviewers will find our responses satisfactory. We are willing to further edit the revised version of the manuscript in response to any additional suggestions they may have.

SPECIFIC COMMENTS

COMMENT: The title is not correct: you speak about the use of PSGs in small animal but in your cases one of the patient is a Golden retriver.

RESPONSE: As suggested, this change has been made (Line 4)

COMMENT: You speak about the use of PSGs to perfrom corrective osteotomy, but in your paper it is included a stifle artrodesis.

RESPONSE:

We agree to the reviewer's comments. But as we mentioned previously, we focus on the evaluate the accuracy in the role of PSGs during surgery. Moreover, It is reported that a closing wedge corrective osteotomy with tarsal arthrodesis was performed in two dogs (Abrams et al. 2020). in addition, corrective osteotomy was primarily performed for tarsal arthrodesis in the patient with malunion in human medicine. Thus, we decided to revise the sentences to avoid misleading as follows (Line 20)

â–¶a stifle arthrodesis -> corrective osteotomy with a stifle arthrodesis

COMMENT: In the title and in all the paper you say that the object of the study is to compare results of novice and experienced surgeon in performing corrective osteotomy, but there is not a statistical analysis in your material and methods selection, so you can't draw and support objectively your results.

RESPONSE: We totally agree to the reviewer's comments that statistical analysis should be included to support my results objectively. Thus, we added a statistical analysis in material and methods. And also, I revised the sentences of this limitation of this study because of the small sample size as follows (Line 250 to 252)

â–¶2.4. Statistical analysis

The statistical analyses were performed using SPSS version 26.0 software (SPSS Inc., Chicago, IL, USA). Errors of translation and rotation were compared between the experienced and novice group. Mann-Whitney test was used to compare significant differences between the groups. Significance was set at p < 0.05.

â–¶ This study has some limitations, including the small sample size of dogs receiving treatment; it is possible that the surgical outcome would be different with a larger sample size.

COMMENT: In the title and in all the paper you say that the object of the study is to compare results of novice and experienced surgeon in performing corrective osteotomy, but in authors contribution there is written that HBL performed the corrective surgery for all dogs. So I do not understand how it is possibile to compare the accuracy between novice and experienced surgeons if all the procedures were carried out by the same surgeon.

RESPONSE: We thank the reviewer for commenting on important issues related to our study. The identity of the primary surgeon was recorded on Table 1. Thus, we revised the sentences as follows (Line 267-268)

â–¶H.B.L. (experienced surgeon) contributed with his broad knowledge of orthopedic and performed the corrective surgery with Y.H.R. (novice surgeon)

COMMENT: Discussion section is quite low of content, with the same concepts repeated more times but not a true and organized discussion has been carried out.

RESPONSE: We agree with the reviewer's comments that the discussion section has quite low content and is needed to be organized. We decided to edit repeated sentences and add new text about PSGs.

â–¶4. Discussion

The postoperative accuracy of PSGs was evaluated using the 2D/3D registration measurement technique. There was no significant difference in the assessment of the postoperative position of the bone between novice and experienced surgeons. Therefore, the results of the present study demonstrate that corrective osteotomy with patient-specific guides can be an effective treatment for even novice surgeons and achieve high owner sat-isfaction.

Small errors after corrective osteotomy may increase the risk of complications in dogs [2,21]. Several factors can affect the errors and results of surgery, namely, preoperative, in-traoperative, and postoperative factors [15]. In this paper, The total average translation was about 0.57, 1.04, and 0.68 mm on the x, y, and z axes, respectively, and average rota-tion, 0.86, 1.59, and 0.89 mm on x, y, and z axes, respectively, when analyzing each de-formity plan. It is likely that the amount of postoperative deviation found in this study could not affect the clinical outcome because all patients show favorable prognoses with-out major complications. Although the errors found in this study (postoperative transla-tion and rotation) appear to be larger than those found in previous research [15], these discrepancies (mm) were minimal compared to the size of the bone. However, the cause of these errors could relate to the procedure while performing an osteotomy or reduction or when inserting a screw. The soft tissue surrounding the bone compromises the surgeon’s vision, and it is difficult to perceive the precise osteotomy line. In addition, bending of the saw and guide due to excessive power from the surgeon can cause postoperative errors. Although the bone loss from osteotomy in 3D rehearsal surgery was calculated as the thickness of a 0.6 mm saw blade (Stryker TPX, Stryker Inc., Kalamazoo, MI, USA), more significant loss of bone from poor handling technique can affect the outcome. However, there were no significant differences between novice and experienced groups (p > 0.05). One of the PSGs advantages is that surgical outcome can be similar to a preoperative plan [7,15]. Therefore, these errors could result from procedures including performing radiog-raphy and comparing the position of bone segments in a computer program.

The PSGs can be categorized by several types depending on their application method [10]. The advantage of PSGs is that various types can be applied by preference of the sur-geon and patient size. The osteotomy guide determines the position of the saw where the surgeon wants without measuring the bone by ruler, increasing the accuracy of designat-ing the osteotomy line and reducing the surgical time [7,14]. All surgeries in this study were completed within two hours regardless of the experience of the surgeon. The 3D-PSG and rehearsal surgery with a bone model helps the surgeon identify the exact location for the osteotomy while allowing the surgeon to be familiar with the bone’s circumstances prior to surgery [8]. For reduction, distracting bones’ fragments in the patients with soft tissue constracture is challenging after osteotomy. In addition, Maintaining the segments of the distracted bone with torsion or varus/valgus deformities is more difficult [10]. The reduction guide not only prevents excessive translation and rotation in the process of re-duction after the osteotomy but also allows segments of bones to be positioned according to plans because the ventral surface of this reduction guide was modeled to completely match the dorsal surface of the postoperatively reduced bone on CAD surgery [7,14]. K-wires for the reduction guide was placed to the bone before making an osteotomy with osteotomy-guide. The position of the k-wire was planned based on the postoperative fea-ture of reduced bone [16]. Therefore, as the reduction guide glide through the k-wires to contact bone segments’ surface, it cause the bone segments to translate and rotate to opti-mal position. Compared to Jig which is commonly used to reduce and stabilize in ortho-pedic surgery, The PSGs are more precise, simple with less iatrogenic bone damage [15]. It could reduce the risk of postoperative bone fracture in small-breed dogs. Although a sig-nificant amount of time is required to prepare PSGs and bone models for rehearsal surgery, this is compensated by improvements in the accuracy of postoperative outcome and shorter operating hours.

This study has some limitations, including the small sample size of dogs receiving treatment; it is possible that the surgical outcome would be different with a larger sample size. Although each case has different guides and different surgical procedures, various types of PSGs could be applied to diverse cases due to their natural and specific features. Moreover, the purpose of this study was not to provide guidelines about the type of PSG that should be clinically indicated in dogs with bone deformities. In addition, the issues of cost and the time for the production of the PSGs must be considered. Further studies are needed to evaluate the efficacy of this technique with regard to complete resolution and the cost of long-term follow-up.

PSGs could be beneficial in surgery, despite the varieties of dog breeds and tech-niques. Although the precise methods and systems for the use of PSGs have not been con-clusively determined, considering the related studies and technological development, the PSG system is thought to be useful for corrective osteotomy on dogs, particularly for nov-ice surgeons who intend to improve their accuracy of surgical correction in veterinary medicine. Based on our experience, the application of PSGs appears to be feasible, result-ing in excellent accuracy and limited postoperative complications.

Round 2

Reviewer 1 Report

the paper appears complete and well written, the idea behind the research is original.

Author Response

Dear Reviewers

We want to thank the reviewers and editor for their thoughtful review of the manuscript. They raise important issues, and their comments have been constructive for improving the manuscript. We hope that the reviewers will find our responses satisfactory. We are willing to further edit the revised version of the manuscript in response to any additional suggestions they may have.

Reviewer 2 Report

I would like to thank the authors for the substantial changes apported to the paper.

line 46: correct pfor with for 
line56-58: rephrase, the sentence is not clear 
line 58-59: there is some problem with spacing 
line 62: full stop is missing at the end of the sentence
line 66: why patients affected by dysplasia were included? 
line 153: replace surgeries with cases 
line 155: replace “in 2 hours” with “within 2 hours”.
line 167: replace ulnar with ulna 
line172-177: not useful, please delete these sentences 
line 191-192: not useful, please delete these sentence
line 193: please report in the text a sentence about the results of statistical analysis 
line 201: change demonstrates with supports

Author Response

Dear Reviewers

We want to thank the reviewers and editor for their thoughtful review of the manuscript. They raise important issues, and their comments have been constructive for improving the manuscript. We hope that the reviewers will find our responses satisfactory. We are willing to further edit the revised version of the manuscript in response to any additional suggestions they may have.

Response to comments from Review 2

COMMENT: line 46: correct pfor with for

RESPONSE: As suggested, this change has been made (Line 46).

COMMENT: line56-58: rephrase, the sentence is not clear

RESPONSE: We would like to thank the reviewer for commenting on important issues that could be misleading. Thus, we revised these sentences as follows (Line 57 to 59)

â–¶ However, there have been a few reports regarding these techniques in veterinary medicine. In addition, these reports have the limitations of a small number of cases to compare.

COMMENT: line 58-59: there is some problem with spacing

RESPONSE: There was some software program related error. As suggested, this change has been made (Line 58 to 59)

COMMENT: line 62: full stop is missing at the end of the sentence

RESPONSE: As suggested, this change has been made (Line 62)

COMMENT: line 66: why patients affected by dysplasia were included?

RESPONSE: We thank the Associate Editor for commenting on important issues related to our study. We agree with the reviewer's comments that there is a clear difference between OA and dysplasia. Thus, we decided to revised these sentences as follows (Line 65 to 66)

â–¶ All patients who underwent a corrective osteotomy with 3D-PSGs for the treatment of bone deformities between 1 January 2018 to 1 June 2020 were eligible for inclusion.

COMMENT: line 153: replace surgeries with cases

RESPONSE: As suggested, this change has been made (Line 62)

COMMENT: line 155: replace “in 2 hours” with “within 2 hours”.

RESPONSE: As suggested, this change has been made (Line 155)

COMMENT: line 167: replace ulnar with ulna

RESPONSE: As suggested, this change has been made (Line 166)

COMMENT: line172-177: not useful, please delete these sentences

RESPONSE: As suggested, this change has been made (Line 172)

COMMENT: line 191-192: not useful, please delete these sentence

RESPONSE: As suggested, this change has been made (Line 187)

COMMENT: line 193: please report in the text a sentence about the results of statistical analysis RESPONSE: We totally agree to the reviewer’s comments that the results of statistical analysis should be inserted. As suggested. We inserted the sentence as follows (Line 182 to 183)

COMMENT: line 201: change demonstrates with supports

RESPONSE: As suggested, this change has been made (Line 196)